# Assessing the Accuracy of Multi-Temporal GlobeLand30 Products in China Using a Spatiotemporal Stratified Sampling Method

Yali Gong [1,2], Huan Xie [1,2,3,*], Shicheng Liao [1,2], Yao Lu [1,2], Yanmin Jin [1,2], Chao Wei [1,2] and Xiaohua Tong [1,2]

1   College of Surveying and Geo-Informatics, Tongji University, Shanghai 200092, China;
    1710643@tongji.edu.cn (Y.G.); 1810647@tongji.edu.cn (S.L.); luyao@alumni.tongji.edu.cn (Y.L.);
    jinyanmin@tongji.edu.cn (Y.J.); cwei@tongji.edu.cn (C.W.); xhtong@tongji.edu.cn (X.T.)
2   Shanghai Key Laboratory of Space Mapping and Remote Sensing for Planetary Exploration, Tongji University,
    Shanghai 200092, China
3   Shanghai Institute of Intelligent Science and Technology, Tongji University, Shanghai 200092, China
*   Correspondence: huanxie@tongji.edu.cn

**Abstract:** The new type of multi-temporal global land use data with multiple classes is able to provide information on both the different land covers and their temporal changes; furthermore, it is able to contribute to many applications, such as those involving global climate and Earth ecosystem analyses. However, the current accuracy assessment methods have two limitations regarding multi-temporal land cover data that have multiple classes. First, multi-temporal land cover uses data from multiple phases, which is time-consuming and inefficient if evaluated one by one. Secondly, the conversion between different land cover classes increases the complexity of the sample stratification, and the assessments with different types of land cover suffer from inefficient sample stratification. In this paper, we propose a spatiotemporal stratified sampling method for stratifying the multi-temporal GlobeLand30 products for China. The changed and unchanged types of each class of data in the three periods are used to obtain a reasonable stratification. Then, the strata labels are simplified by using binary coding, i.e., a 1 or 0 representing a specified class or a nonspecified class, to improve the efficiency of the stratification. Additionally, the stratified sample size is determined by the combination of proportional allocation and empirical evaluation. The experimental results show that spatiotemporal stratified sampling is beneficial for increasing the sample size of the "change" strata for multi-temporal data and can evaluate not only the accuracy and area of the data in a single data but also the accuracy and area of the data in a multi-period change type and an unchanged type. This work also provides a good reference for the assessment of multi-temporal data with multiple classes.

**Keywords:** accuracy assessment; spatiotemporal stratified random sampling; area estimation; GlobeLand30

## 1. Introduction

Land cover and land cover change data are essential basic information and key parameters for global climate change, the ecological assessment of natural resources, and the monitoring of geographical conditions [1,2]. Moreover, they help to determine water balance, the carbon cycle, and energy exchange [3–5]. Land cover change is an essential prerequisite for the implementation of global initiatives in sustainable development and for the preservation of biodiversity and ecosystem functions [6]. Land cover is also a direct reflection of the interaction between human activities and nature. For example, the spatial distribution and pattern of land cover also affect economic conditions, health, and wealth [2,7,8]. Providing timely and effective information pertaining to global, national, and regional data on land cover and land use change can be important. Maps for different time periods usually provide the basis for monitoring changes in land cover, and they can

take advantage of a sampling approach to land cover change monitoring. A sampling-based assessment provides uncertainty in terms of the data product, which is very important in the development and application of remote sensing technology. Firstly, the analysis of the accuracy assessment will enhance the confidence of the relevant applications using remote sensing data products. Then, remote sensing products can be a reliable information source to support many types of applications at the global, national, and regional scales, such as global climate change, the ecological assessment of natural resources, and the monitoring of geographical conditions challenges, such as global change [9–11]. Secondly, the accuracy assessment indicates various sources of errors in the remote sensing data products, which can provide good feedback for remote sensing technology. Based on accuracy assessments, remote sensing technology can be further improved.

Accuracy verification is a crucial technique for assessing the accuracy and area of high-resolution land cover data at both the global and regional scales [12]. A robust and statistically evaluated map serves as an estimation of classification accuracy, and it is expressed in an overall and per-category design-based inference [2,13,14]. Sampling-based accuracy assessments extend to area estimation, which has a more direct economic impact on land cover, and these areas can range from cropland, forest, artificial surfaces, etc. Due to classification errors, most pixels that change attribute labels correspond to classification errors rather than land cover changes, and area estimations that use a direct pixel-based counting method are biased [15]. The good practice guidelines emphasize that land cover data and multi-temporal land cover data should be estimated based on reference data such that the estimates are unbiased [16].

Land cover data products provide a strata basis for post-hierarchical estimations, which are used to reduce standard errors. At present, domestic and foreign scholars mainly study the accuracy estimation methods of multiclass land cover data in single data and multi-temporal single-class land cover data. There are few studies on how to reasonably stratify multi-temporal and multiclass land cover data. Tang presented a framework with which to assess the accuracy of the near real-time monitoring of tropical forest disturbances, which was based on stratified sampling, and this was achieved by combining three mapped datasets for 2013, 2014, and 2015 (which were produced by Fusion2, near real-time continuous-change detection and classification (NRT-CCDC), and Terra-I, respectively [4]). Wickham verified the accuracy of NLCD2016 and NLCD2019 land cover and evaluated all of the categories of unchanged and changed strata together [17,18]. The original changed strata did not contain the content of the later changed strata so as to achieve the purpose of effective stratification [17,18]. Arévalo stratified six stable land cover strata, five dynamic land change strata (which represented the 2001–2016 period), and one buffer stratum to reduce the impact of the missing errors in the continuous monitoring of land change activities and the postdisturbance dynamic tests of Landsat time series [19].

Although these products have been validated for accuracy using various methods, the methods employed are not suitable for multi-temporal and multiclass land cover data products. Due to the rapid accumulation of multi-temporal and multiclass land cover data, the amount of data is large, and the spatiotemporal variation in land cover classes is complex. Multi-temporal land cover can provide data support for accuracy and for the area estimation of changing ground object types over at least three time periods. The quality of classifiers comes from the map production process, while the effectiveness of stratification is highlighted in the accuracy evaluation process. For multi-temporal and multifactor land cover data, their effective stratification directly affects the accuracy of accuracy estimations and area estimations. There are still some bottlenecks for the accuracy evaluation of multi-temporal data. The conversion between different land covers will improve the complexity of the sample stratification. An assessment with different types of land cover will suffer from inefficient sample stratification since the multi-temporal and multiclass land cover data have a mutual transformation of multiple land cover categories in multiple phases. If the multi-temporal and multiclass land cover data are to be divided into multiple strata that do not overlap spatially, each transformation type must be set as a separate stratum

such that there will be hundreds of strata, thousands of strata, or even greater amounts of strata (which is not suitable for the stratified sampling method). Second, the qualitative analysis of land cover change is particularly important for practical applications, but the area proportion of a stable class is larger than that of a changed class, which will lead to inappropriate sample allocation. Therefore, it is urgent to study the above problems and understand how to reasonably stratify multi-temporal and multifactor land cover data. In this paper, we propose a spatiotemporal stratified sampling approach for multi-temporal data with multiple classes. In this method, the data of each class of land cover in each period are extracted separately, and the data of a certain class of land cover in multiple periods are combined. Then, the spatiotemporal stratification is carried out according to the combination of the change and the unchanged types of land cover class in multiple phases. In order to facilitate statistics and to improve the efficiency of the stratification, the specific conversion relationship between the types of land cover was not considered. The temporal changes in single-class data and the spatial location of different classes were used to obtain a reasonable stratification. When compared with stable unchanged strata, the changed strata account for a small proportion and are prone to classification errors.

The research objective of this paper is to test the proposed spatiotemporal stratified sampling method, which is based on single-class, spatiotemporal changed and unchanged types combined with the estimation protocol, and we apply this to the GlobeLand30 China [20] region from 2000 to 2020. This paper quantitatively describes the land cover changes and non-changes in China over a 20-year period, which includes the following: (1) evaluating the accuracy of China land cover data in 2000, 2010, and 2020, and (2) estimating the accuracy and area of land cover change and non-change in China over the last 20 years.

## 2. Methodology

In this work, a statistical inference-based sampling method was utilized to independently stratify each land cover class, and the area and accuracy of each land cover class with and without change were accurately evaluated. The accuracy estimation and area estimation of GlobeLand30 data in China followed good practice recommendations [16]. The experimental process in this paper is shown in Figure 1. The experimental data were preprocessed to generate the GlobeLand30 data for the three periods in China, and then the spatially representative sample pixels were selected based on the spatiotemporal stratified random sampling method for visual interpretation. This provided the experimental preprocessed data from the years 2000, 2010, and 2020 for the GlobeLand30 data in China. The different classes of land cover in the three periods were divided into three to seven strata according to the importance of the change type conversion from 2000 to 2020 [16,21]. The stratified sampling that is based on the combination of changed and unchanged types of land cover classes in multiple phases can ensure that rare strata have enough samples to achieve the purpose of accurately estimating its user accuracy [18]. The total sample size of each class was calculated based on the probabilistic statistical sampling optimization model [22]. A 30 m resolution pixel of each class served as the sampling unit. Based on a probabilistic sampling design, stratified random sampling was employed for sample selection in each spatiotemporal stratum. The visual interpretation of the reference data was conducted to quantify the parameter estimations that were based on the samples [21,23]. Finally, the approach for assessing land cover accuracy was described, and the accuracy of the results was analyzed.

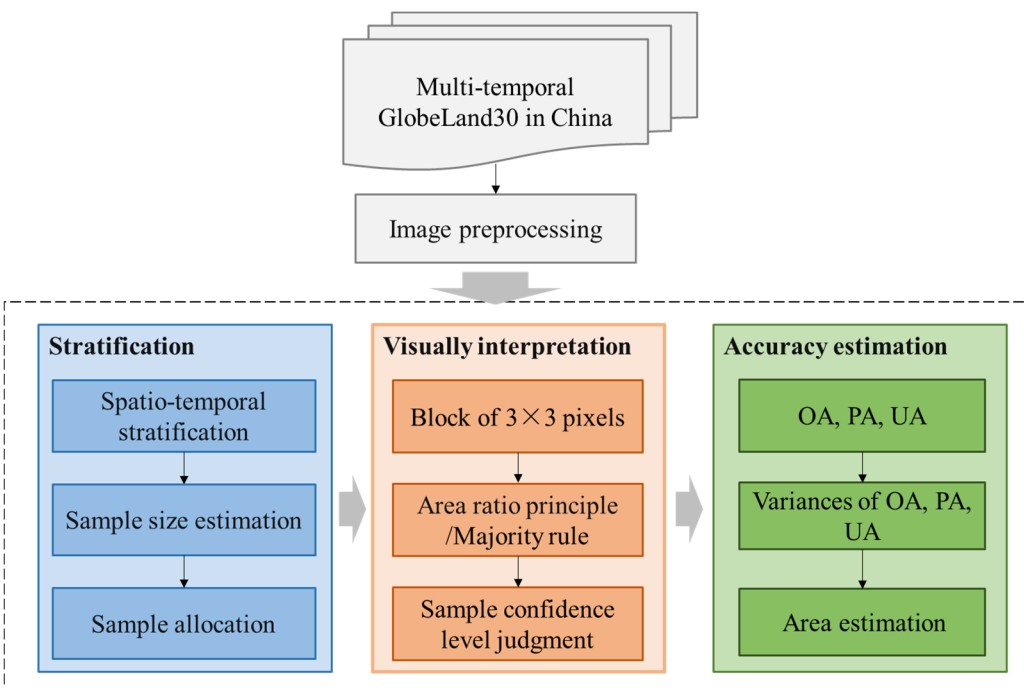

**Figure 1.** The overall workflow for the accuracy assessment of multi-temporal and multiclass GlobeLand30 data in China.

*2.1. Dataset*

The GlobeLand30 dataset contains rich and detailed information about the spatial distribution of land cover, which can reflect human land use activities and the landscape patterns formed by them. In this experiment, the Chinese dataset from 2000 to 2020 was obtained from the Welcome-GlobeLand30 website (globeland30.org (accessed on 2 August 2023)), and the images of the same year were imported into ArcMap 10.6. Since multiple images were downloaded, a series of data preprocessing was required to generate the entire land cover data of China. The black edges between the images were eliminated, and all images were embedded into one. Through the boundary of China, shp data were clipped into the mosaic image. The GlobeLand30 data for the China region were obtained, as shown in Figure 2. For a better spatial analysis of the data, the data projection was converted from D_WGS_1984 to GCS_WGS_1984. The GlobeLand30 data in China mainly include nine land cover classes, i.e., cultivated land, forest, grassland, shrub, wetland, water, artificial surfaces, bare land, and ice and snow [20], as shown in Table 1. The publication of GlobeLand30 data in 2000, 2010, and 2020 provided important data support for the regional land cover change evaluation in China from 2000 to 2020. Since the beginning of the 21st century, with the acceleration of industrialization and urbanization, land cover types have changed significantly in China [24]. Although the significance of land cover change is well recognized by researchers, there remains a dearth of quantitative analysis on this topic in China, and this impedes the widespread utilization of land cover datasets.

**Table 1.** GlobeLand30 land cover classes [20].

| Category | Description |
| --- | --- |
| Cropland | Area used for the production of annual cultivated crops, such as corn, paddy land, vegetables, fruit trees, tilled tidal flats, mudflats, etc. |
| Forest | Area refers to land with a crown density of more than 10% but also includes land with a crown density of less than 10% that is not used for other land types. |

**Table 1.** *Cont.* [20].

| Category | Description |
|---|---|
| Grassland | Area dominated by natural grassland with a total vegetation coverage ratio of more than 20%, including areas used for animal husbandry production and all kinds of natural grassland, such as meadows, savannas, etc. |
| Shrub | Area dominated by relatively low-growing plants without a main stem and with a total vegetation cover of more than 20%. |
| Wetland | A confluence of land and water near or at the surface of the ground or of shallow water and soil with a bog or hygrophyte growth in a wetland section. |
| Water | Liquid water on land surfaces, including rivers, lakes, reservoirs, ponds, fishponds, etc., except for cultivated land, such as paddy fields, wetlands, multi-year snow-covered areas or glaciers, and marine types. |
| Artificial surfaces | Area formed by artificial activities covered by asphalt, concrete, sand, stone, brick, glass, and other building materials, including residential areas, industrial and mining land, land for transportation facilities, etc. |
| Bare land | Land with less than 20% total vegetation cover, including saline-alkali surfaces, sandy land, gravel land, rocky land, biological crusts, etc., but excludes human-made cover, tidal flats, sea surfaces, etc. |
| Ice and snow | Permanent snow-covered areas, ice caps, and glaciers. Seasonal snow and ice-covered areas on land and water are not included in this category. |

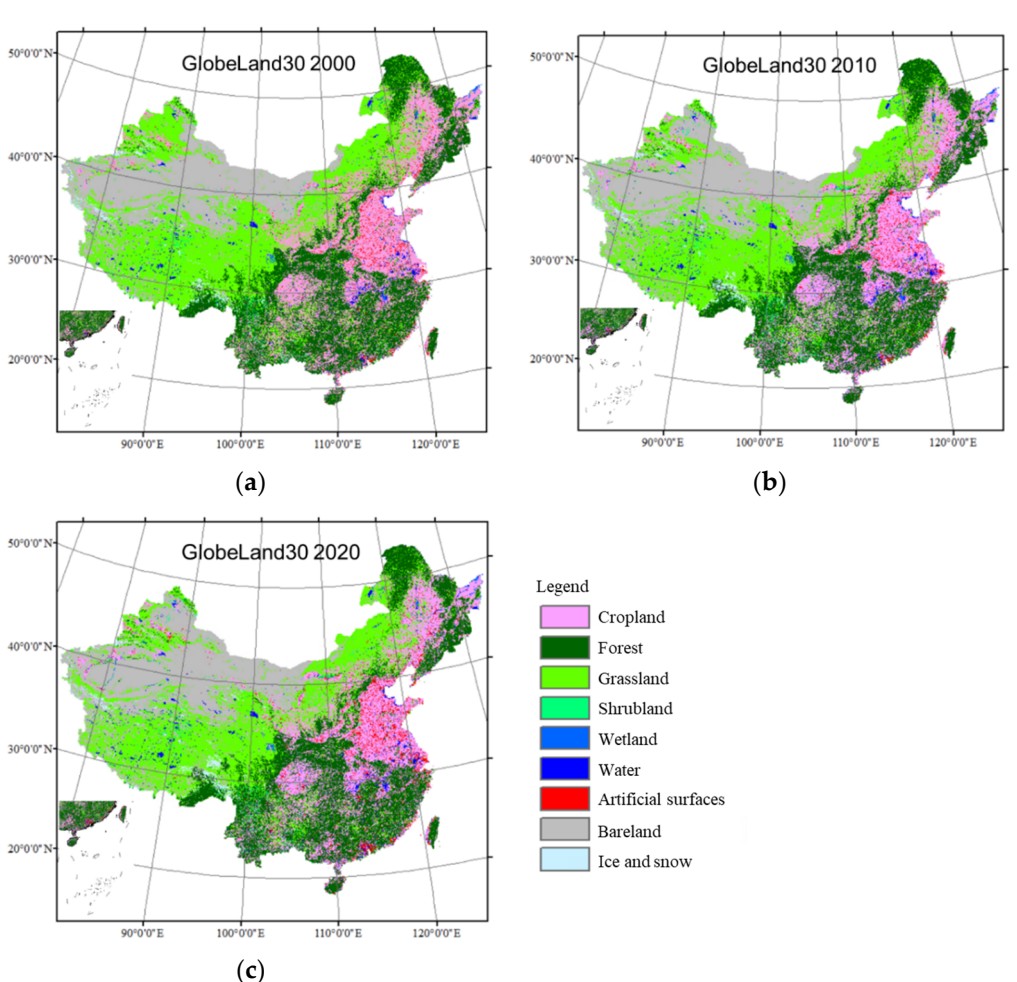

**Figure 2.** GlobeLand30 data in China for (**a**) 2000, (**b**) 2010, and (**c**) 2020.

## 2.2. Spatiotemporal Stratification and Simplification

The purpose of the accuracy assessment of multi-temporal GlobeLand30 data for China is to determine the quality of the data in China by statistical inference via samples that are representative of the overall quality of the data. In contrast to the traditional stratification method that is based on land cover categories, the increase in temporal domains significantly improves the complexity of data product quality assessment. How to reasonably stratify land cover data is the focus of this research. In this experiment, there are evident differences between the pixel change and the unchanged data of the three periods of GlobeLand30 data in China. Thus, the stratification of the strata according to individual categories of spatiotemporal changed and unchanged information ensures that the datasets are distributed in each spatiotemporal distribution context and that they can be evaluated. In order to assess the accuracy and the proportion of the area of the GlobeLand30 data in China, the changed and unchanged types of each class of land cover for the three periods were used as the stratification basis [25,26]. The spatial–temporal stratified sampling that is proposed in this study, as shown in Figure 3, is as follows:

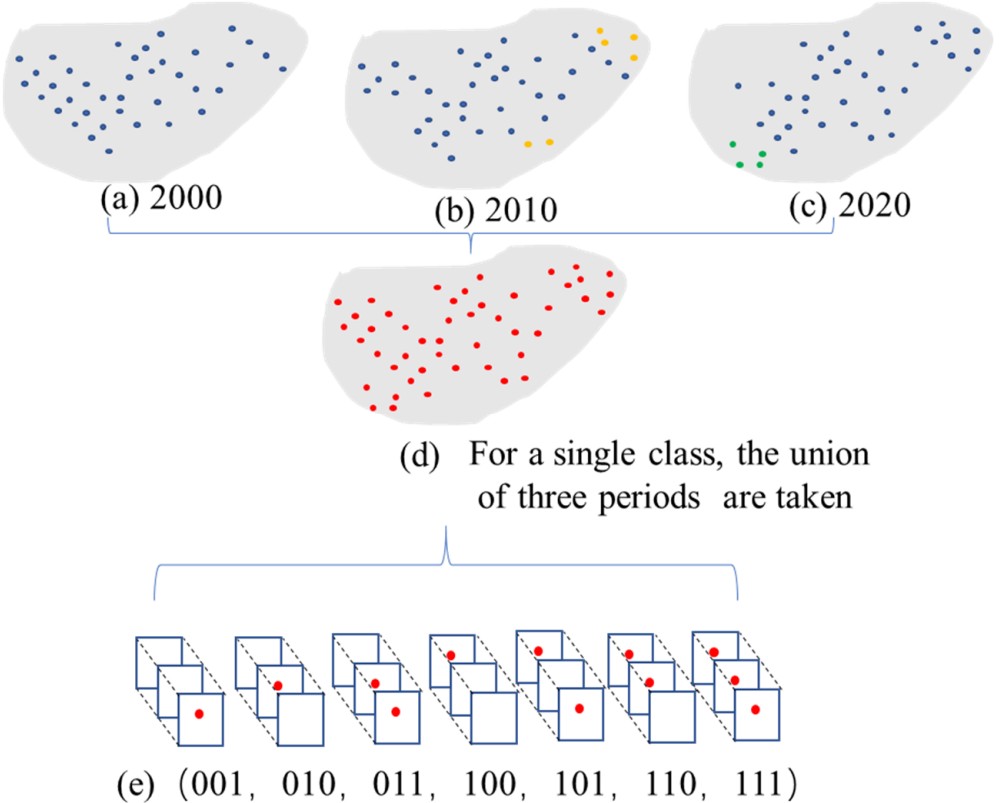

**Figure 3.** The flowchart of spatiotemporal stratified sampling. (**a**) Land cover class in 2000, as shown by the blue dots; (**b**) land cover class in 2010 (yellow dots indicate the new pixels of the similar classes when compared to 2000); (**c**) land cover in 2020 (green dots indicate the new pixels of the similar classes when compared to 2010); (**d**) the three periods of land cover taken and combined together, as shown by the red dots, and (**e**) the spatiotemporal stratified sampling. (001–111 is the binary code of the spatiotemporal stratification of each land cover class corresponding to Table 2).

**Table 2.** Temporal and spatial stratification of the multi-temporal land cover data.

| Category | Stratum | Binary Coding |
|---|---|---|
| Cropland | Stable cropland, cropland gain from 2000 to 2010, cropland loss from 2000 to 2010, cropland gain from 2010 to 2020, cropland loss from 2010 to 2020, cropland gain then loss, cropland loss then gain. | 111, 011, 100, 001, 110, 010, 101 |
| Forest | Stable forest, forest gain from 2000 to 2010, forest loss from 2000 to 2010, forest gain from 2010 to 2020, forest loss from 2010 to 2020, forest gain then loss, forest loss then gain. | 111, 011, 100, 001, 110, 010, 101 |
| Grass | Stable grass, grass gain from 2000 to 2010, grass loss from 2000 to 2010, grass gain from 2010 to 2020, grass loss from 2010 to 2020, grass gain then loss, grass loss then gain. | 111, 011, 100, 001, 110, 010, 101 |
| Shrub | Stable shrub, shrub gain from 2000 to 2010, shrub loss from 2000 to 2010, shrub gain from 2010 to 2020, shrub loss from 2010 to 2020, shrub gain then loss, shrub loss then gain. | 111, 011, 100, 001, 110, 010, 101 |
| Wetland | Stable wetland, wetland loss from 2000 to 2010, wetland loss from 2010 to 2020, wetland gain from 2000 to 2010, wetland gain from 2010 to 2020. | 111, 011, 100, 001, 110 |
| Water | Stable water, water gain from 2000 to 2010, water loss from 2000 to 2010, water gain from 2010 to 2020, water loss from 2010 to 2020, water gain then loss, water loss then gain. | 111, 011, 100, 001, 110, 010, 101 |
| Artificial surfaces | Stable artificial surfaces, artificial surfaces gain from 2000 to 2010, and artificial surfaces gain from 2010 to 2020. | 111, 011, 001, |
| Bare land | Stable bare land, bare land gain from 2000 to 2010, bare land loss from 2000 to 2010, bare land gain from 2010 to 2020, bare land loss from 2010 to 2020. | 111, 011, 100, 001, 110 |
| Ice and snow | Stable ice and snow, ice and snow gain from 2000 to 2010, ice and snow loss from 2000 to 2010, ice and snow gain from 2010 to 2020, ice and snow loss from 2010 to 2020. | 111, 011, 100, 001, 110 |

Firstly, the multi-temporal and multiclass land cover data are listed separately for each class in each period (Figure 3a–c), and the three-phase data of the same land category are combined as candidate samples (Figure 3d). The GlobeLand30 data in China with nine land cover categories were divided into nine types of sample candidate regions.

Secondly, spatiotemporal stratified sampling was conducted through a selection of the changed and unchanged types of each class of data in the three periods (Figure 3e). In the stratification process of this experiment, only the changed and unchanged type of a certain category during the 2000–2020 period is considered. It is impossible to collect a sufficient sample size to estimate the area of a specified class that is converted to other classes at an acceptable level of cost and precision [27]. Therefore, the labeled combination of changed and unchanged data in a single category over multiple periods was the basis for stratification.

In order to facilitate the recording, the specified class and nonspecified class data were labeled using binary coding, i.e., 0 and 1, where 1 represents the specified class, and 0 represents the nonspecified class data. In certain regions of China, the changed type of land cover class occupies a small proportion such that there are only a few or no sample pixels for it. When considering the change in area proportion, spatiotemporal stratification follows the rule regarding classes where the priorities change or have no change [28]. Therefore, this small amount of changed land cover type class of information

is not taken as a stratum in this experiment. The spatiotemporal stratification of the nine land cover classes in the GlobeLand30 data of China is shown in Table 2. In the joint stratification of multiple land cover classes, other classes of data compared to one class of data represent examples of nonspecific types of data; as such, stable nonspecified classes were no longer regarded as separate strata in the process of the spatiotemporal stratification of multi-temporal and multiclass land cover. This stratified approach is also different from the previous stratified sampling design of multi-temporal single-class data [6,29,30]. In this experiment, the spatiotemporal stratified random sampling method was adopted to select sample pixels, and this method can ensure that there are enough samples of the interested changed stratum for sample-based statistical estimations [4,16].

### 2.3. Sample Size Determination and Allocation

The sample assessment unit in this study is a single pixel at a 30 m spatial resolution, and these units were selected from each stratum and were consistent with Strahler's [31] "best practice" recommendations. The sample size of each class was calculated based on the probability statistical sampling model [23]. Then, the sample size for each stratum of each class was based on a proportional allocation. In the case of determining the total sample size, the sample size of each class is allocated according to the proportion of its area to the total area. Generally, changed strata may be assigned a small sample size due to the small, mapped area, which makes accurate estimates of user accuracy for changed strata impossible. To solve the problem, the sample size of these strata should be increased. If the sample size allocated to the rare strata is small, it can be filled up to 100. The 100 sample pixels in certain strata introduced trade-offs between the interpretation work and an acceptable margin of error in the estimation [32]. The number 100 is actually an empirical value, which was obtained from reference [16]. The previous research shows that such a sample size performs better than other allocations, showing minimal estimated variances in the user's accuracy, producer's accuracy, and the proportion of area. Since the area proportion of the changed strata is often smaller than other strata, setting a minimal sample size is an important way to get a reliable assessment. Then, the sample pixels were randomly selected from all pixels that were mapped as that type in the stratum.

### 2.4. Accuracy Analysis and Area Estimation

Accuracy analysis is the statistical estimation of the results following the visual interpretation of samples obtained through probability sampling [33,34]. The results of visual interpretation can be presented as error matrices, where the rows represent the map labels and the columns represent the reference labels [12]. The protocol of agreement is defined as a match between the map label and the reference label [35]. The reference data are high-resolution Google Earth images [36,37], and each sample pixel is visually interpreted by two experts independently. For pixels with different interpretation results, three or more experts collectively discuss them, and consistent attribute labels are given according to the majority rule [38]. The analyses are based on reference labels for three dates: 2000, 2010, and 2020, with a resolution of 30 m per sample pixel. Enlarging each sample pixel point to a block of 3 × 3 pixels can effectively prevent the impact caused by the inherent position error and geographic matching error of satellite imaging [39]. The correct interpretation of pixels is performed by formulating judgment principles, such as effective area proportion and the percentage of pixels matched correctly [40]. The Google Earth images obtained for each sample pixel in 2000, 2010, and 2020 were visually judged. The slider time toolbar helps the interpreter view images of multiple dates in the same sample location [27]. When the time period corresponding to the Google Earth high-resolution image does not conform to the time period of the sample pixels, it can be judged based on the images about 2 years before and after the same period of the sample or from the combination of multiple time series images. The recording of the land cover class composition of each sample pixel and the reference labels were provided by interpreters. If the reference label of the sample pixel was stable over the three periods, then it was considered unchanged. In the visual

interpretation of whether the sample pixels were considered as having changed, the label of pixel *i* in the previous year could not be equal to the label of pixel *i* in the next year.

For the accuracy estimation of multiclass data in a single-year period and the accuracy estimation of changed and unchanged types between two periods, the strata and map classes were different; thus, it was reasonable to use Stehman's [12] stratified estimator. Each cell of the error matrix represents the estimated percent of an area for each defined class as a match between the map and reference label [17,33]. The following formula was used to estimate the accuracy of the agreement measures and the associated standard errors. The overall accuracy (OA) is defined as follows [12,18]:

$$\hat{OA} = (\frac{1}{N}) \sum_{h=1}^{H} N_h \hat{P}_h \tag{1}$$

where $\hat{P}_h = \sum_{u \in h} \frac{y_u}{n_h}$ is the sample proportion of the correctly classified pixels in stratum *h* ($y_u = 1$ if pixel *u* is classified correctly; otherwise, $y_u = 0$). Furthermore, $u \in h$ implies that sample pixel *u* was selected from stratum *h*. $n_h$ is the sample size in stratum *h*, and *N* is the total number of pixels in the region of interest, whereas $N_h$ is the number of pixels in stratum *h*. *H* is the population size of the strata. The variance of OA is defined as [12,18]

$$\hat{V}(\hat{OA}) = (\frac{1}{N^2}) \sum_{h=1}^{H} N_h^2 (1 - \frac{n_h}{N_h}) s_{yh}^2 / n_h \tag{2}$$

where, $s_{yh}^2$ is the sample variance of $y_u$ within stratum *h*.

The overall accuracy was estimated for the land cover products of three time periods, and this was achieved by taking into account the specific types of the binary map of changed and non-changed land cover. For example, the overall accuracy of the types of change and unchanged cropland from 2000 to 2010. This gives the overall accuracy of the combination of the changed and unchanged types of cropland over three time periods: 2000–2010–2020.

The user accuracy (UA) and producer accuracy (PA) are the empirical estimates of conditional probabilities P (the reference label of class *A* | mapped as class *A*) and P (mapped as class *A* | the reference label of class *A*). In the practical application, the two indicators were estimated as a notification ratio [12]:

$$R = \frac{Y}{X} \tag{3}$$

where *Y* is the population total of $y_u$, which is defined as follows:

$$y_u = \begin{cases} 1, u \in \text{condition} A \\ 0, u \notin \text{condition} A \end{cases} \tag{4}$$

*X* is the population total of $x_u$, which is defined as follows:

$$x_u = \begin{cases} 1, u \in \text{condition} B \\ 0, u \notin \text{condition} B \end{cases} \tag{5}$$

For example, to estimate the user accuracy of cropland, *A* is the pixel *u*, which is mapped as cropland, and the label of the reference data is also cropland. *B* would be pixel *u*, which is mapped as cropland. If the producer accuracy of the cropland is estimated, *A* is the pixel *u*, which is mapped as cropland, and the label of the reference data is also cropland. *B* would then be the pixel *u*, which has a reference label of cropland. The ratio can then be reformed into the following [12]:

$$\hat{R} = \frac{\hat{Y}}{\hat{X}} = \frac{\sum_{h=1}^{H} N_h \bar{y}_h}{\sum_{h=1}^{H} N_h \bar{x}_h} \tag{6}$$

where $\bar{x}_h$ is the sample mean of $x_u$ for stratum $h$, and $\bar{y}_h$ is the sample mean of $y_u$ for stratum $h$. The variance of the ratio is estimated as follows:

$$\hat{V}(\hat{R}) = \left(\frac{1}{\hat{X}^2}\right)\left[\sum_{h=1}^{H} N_h^2(1 - n_h/N_h)(s_{yh}^2 + \hat{R}^2 s_{xh}^2 - 2\hat{R}s_{xyh})/n_h\right] \tag{7}$$

where $n_h$ is the sample size in stratum $h$, $s_{xh}^2$ is the sample variance of $x_u$ for stratum $h$, $s_{yh}^2$ is the sample variance of $y_u$ for stratum $h$, and $s_{xyh}$ is the sample covariance of $x_u$ and $y_u$ in stratum $h$.

$$s_{yh}^2 = \frac{1}{n_h-1}\sum_{i=1}^{H}(y_u - \bar{y}_h)^2$$
$$s_{xh}^2 = \frac{1}{n_h-1}\sum_{i=1}^{H}(x_u - \bar{x}_h)^2 \tag{8}$$
$$s_{xyh} = \sum_{i=1}^{H}(y_u - \bar{y}_h)(x_u - \bar{x}_h)/(n_h - 1)$$

Since the stratification does not correspond to land cover classes, the user accuracy and the producer accuracy of the changed type should be calculated using the samples from multiple classes [27]. For example, the samples used for estimating the user accuracy of changed cropland from 2000 to 2010 include two types. The first is those samples mapped as cropland in 2000 that were changed to other classes in 2010. The other is those samples mapped as other classes in 2000 that were changed to cropland in 2010. If there is no sample pixel in a stratum satisfying condition A, the parameters are set as $y_u = 0$, $\bar{y}_h = 0$ and the sample variances ($s_{yh}^2$) are equal to 0. If there is no sample pixel in a stratum satisfying condition B, the corresponding parameters are set as $x_u = 0$, $\bar{x}_h = 0$ and the sample variances ($s_{xh}^2$) are equal to 0.

To estimate the proportion of the area of the 2000–2010 and 2010–2020 land cover changed for each class, e.g., gain, loss, and net, we defined $y_u$ as follows:

$$y_u = \begin{cases} 0 \to 1 & u\text{changed to the target class (gain)} \\ 0 \to 0, 1 \to 1 & u\text{did not change} \\ 1 \to 0 & u\text{changed from the target class (loss)} \end{cases} \tag{9}$$

The estimated area of change for a single class in square kilometers was calculated as follows:

$$\hat{A} = 0.0009 * N \sum_{u=1}^{n} y_u/n \tag{10}$$

where $n$ is the total sample size. The standard error of the area estimate was calculated as follows [25]:

$$SE(\hat{A}) = 0.0009 * Ns/\sqrt{n} \tag{11}$$

where $s = \sqrt{\sum_{u=1}^{n}\frac{(y_u-\bar{y})^2}{n-1}}$ is the standard error of sample.

## 3. Results and Discussion

Based on the results of the visual interpretation of the reference sample, the corresponding error matrix was obtained, and the accuracy and area of the classes were further estimated [32]. In this study, we used the three periods of the GlobeLand30 data for China to verify whether the spatiotemporal stratified sampling method is suitable for multi-temporal land cover data. The spatiotemporal stratified sampling design was based on the changed and unchanged types of each class in the three periods. This approach can estimate the accuracy of singular data, binary change, and the three period changes of land cover data, which includes the overall accuracy, user accuracy, and producer accuracy with the corresponding standard error. The area of each land cover area class can be calculated

by the error matrix, including the singular data area of land cover, the area of changes between two phases, and the corresponding estimated standard errors [24,32,41].

### 3.1. The Spatiotemporal Stratified Sampling of GlobeLand30 in China

The purpose of spatiotemporal stratified sampling was to obtain a sample subset of the three periods of the GlobeLand30 data for China. First, the union of the three periods of the individual land cover classes of the GlobeLand30 data for China was extracted. Second, each land cover class was divided into several strata by using a combination of the changed and unchanged types of each class in the three periods (Figure 4). For example, when the three periods showed a change in China's regional forest—on the basis of spatial and temporal changed and unchanged forest in the three periods being classified into seven strata (as shown in Figure 4b)—these were, respectively, the following: the stable forest stratum (111), which represents forest from 2000 to 2020, the forest loss stratum (100, 110), which could be divided into two cases: forest in 2000 and non-forest in 2010 or forest in 2000 but non-forest in 2020. The forest gain stratum (001 and 011) was divided into two cases: non-forest in 2000 and forest in 2010 or non-forest in 2000 and forest in 2020. The forest dynamic changed stratum (010 and 101) refers to the non-forest areas in 2000, the forest areas in 2010, and the non-forest areas in 2020, as well as the forest areas in 2000, the non-forest areas in 2010, and the forest areas in 2020. The sample size of each land cover class was determined according to the probability statistical sampling model [23] (Table 3). A total of 9338 samples were selected for accuracy assessment, with each sample spatial location corresponding to three periods of reference labels. The sample size of each stratum was allocated in proportion to the area. The strata for which the sample size was less than 100 were adjusted to 100. The distribution of the forest samples in different strata was taken as an example (Table 4). According to the area ratio of each stratum pixel-to-total pixel, the sample size allocated to the forest gain then loss stratum, the forest loss during the 2000–2010 period stratum, and the forest loss then gain stratum were less than 100. Then, the sample size of these three strata was adjusted to 100. Finally, the sample size allocated to each stratum of the forest is shown in the adjusted results of Table 4. The sample pixels of each stratum were randomly selected for inspection, and their spatial distribution is shown in Figure 5.

**Table 3.** Sample size of each land cover class.

| Land Cover Class | Sample Size | Land Cover Class | Sample Size |
|:---:|:---:|:---:|:---:|
| Cropland | 1987 | Water | 1030 |
| Forest | 1236 | Artificial surfaces | 622 |
| Grassland | 1041 | Bare land | 1107 |
| Shrub | 717 | Ice and snow | 904 |
| Wetland | 694 | Total | 9338 |

**Table 4.** Sample size of each stratum of forest in the three periods.

| Stratum | Pixels | % | Proportionally | Adjusted |
|:---:|:---:|:---:|:---:|:---:|
| Forest gain from 2010 to 2020 | 74,895,126 | 6.5 | 143 | 143 |
| Forest gain then loss | 36,502,895 | 3.1 | 70 | 100 |
| Forest gain from 2000 to 2010 | 52,240,761 | 4.5 | 100 | 100 |
| Forest loss from 2000 to 2010 | 46,593,596 | 4 | 89 | 100 |
| Forest loss then gain | 31,190,465 | 2.7 | 59 | 100 |
| Forest loss from 2010 to 2020 | 60,876,520 | 5.3 | 116 | 116 |
| Stable forest | 857,387,729 | 73.9 | 557 | 557 |
| Total | 1,159,687,092 | 100% | 1134 | 1216 |

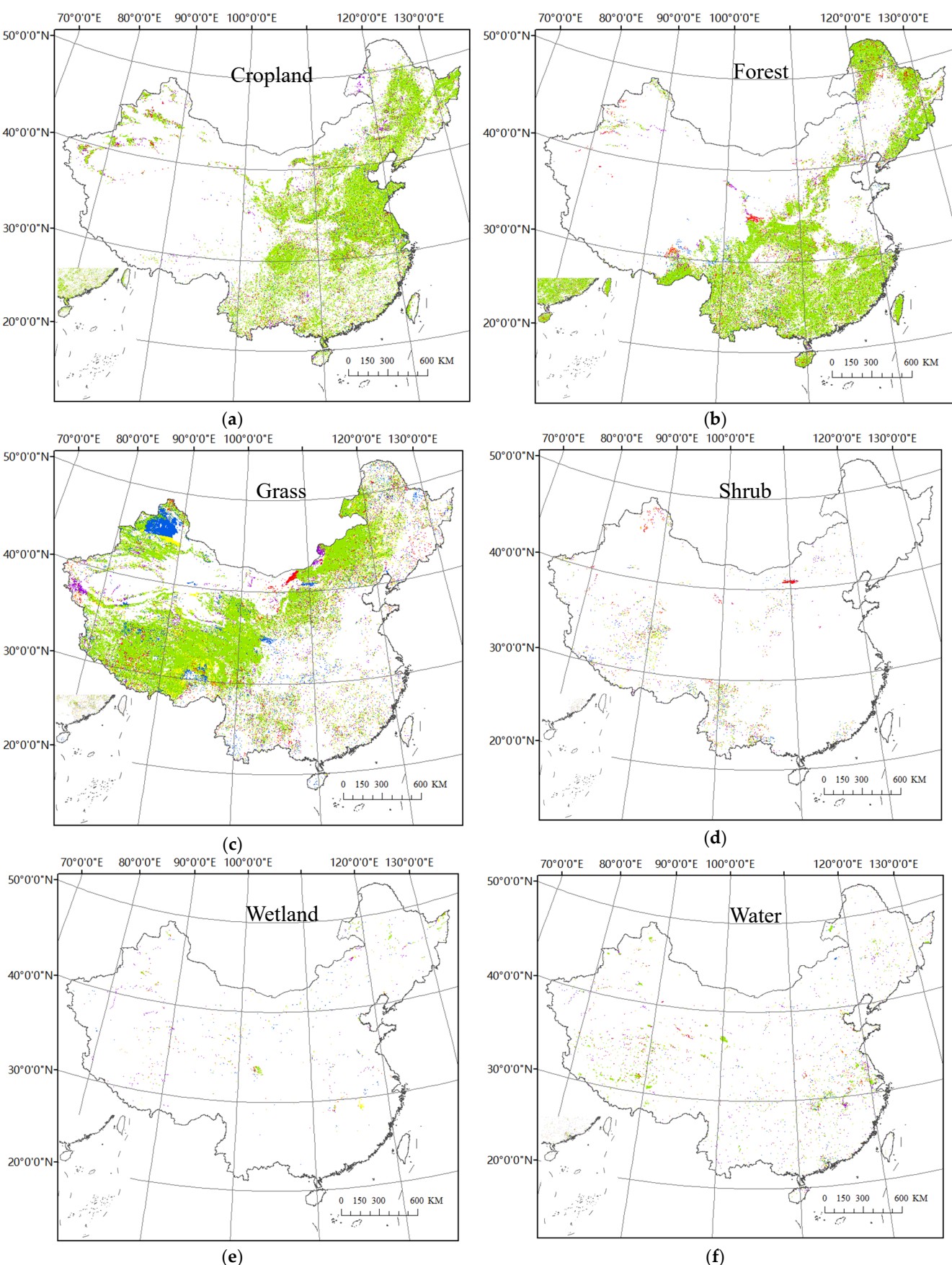

**Figure 4.** *Cont.*

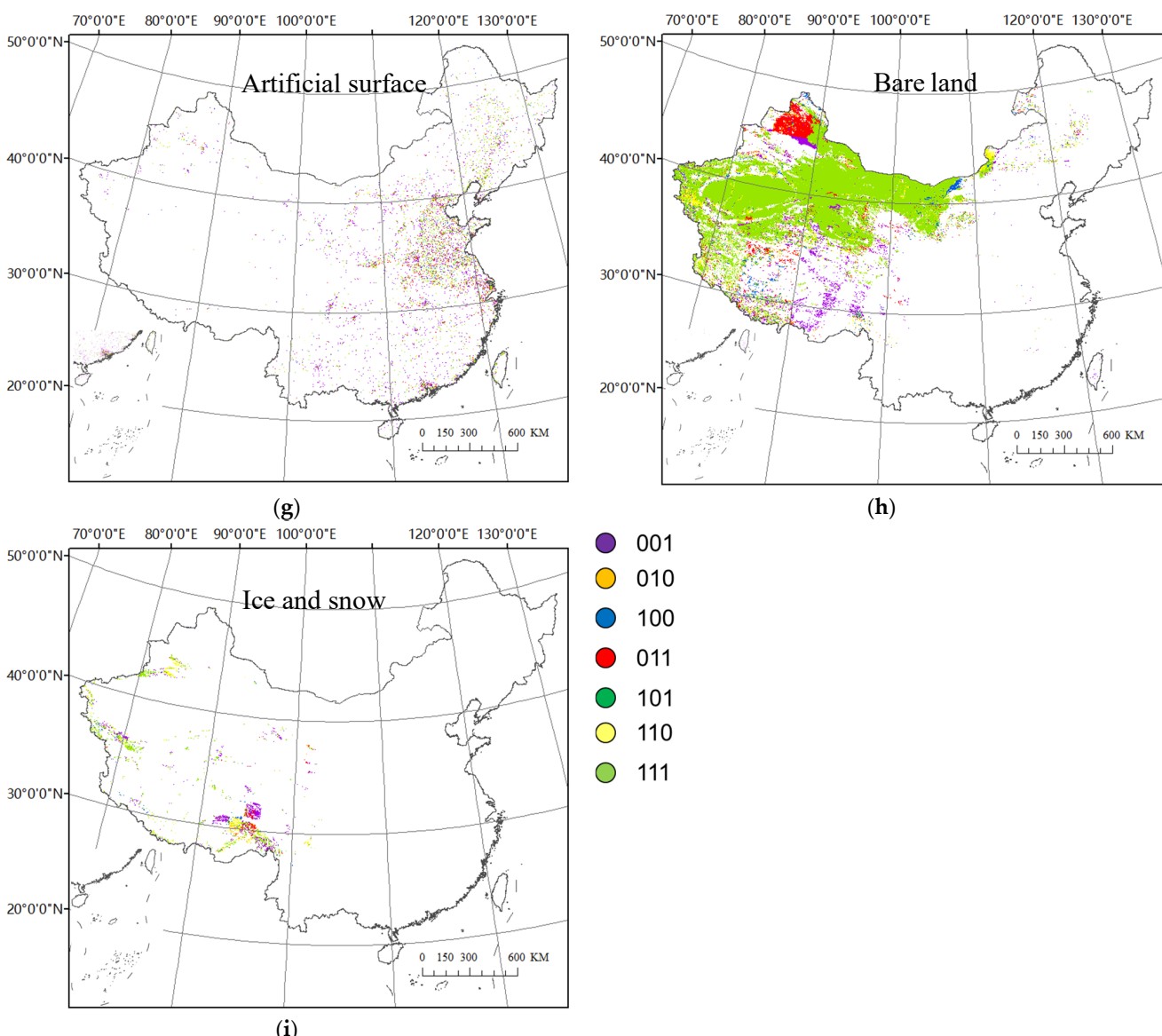

**Figure 4.** Spatiotemporal stratification of each class (001–111 is the binary code for the spatiotemporal stratification for each land cover class corresponding to Table 2). (**a**) Cropland; (**b**) Forest; (**c**) Grass; (**d**) Shrub; (**e**) Wetland; (**f**) Water; (**g**) Artificial surface; (**h**) Bare land; (**i**) Ice and snow.

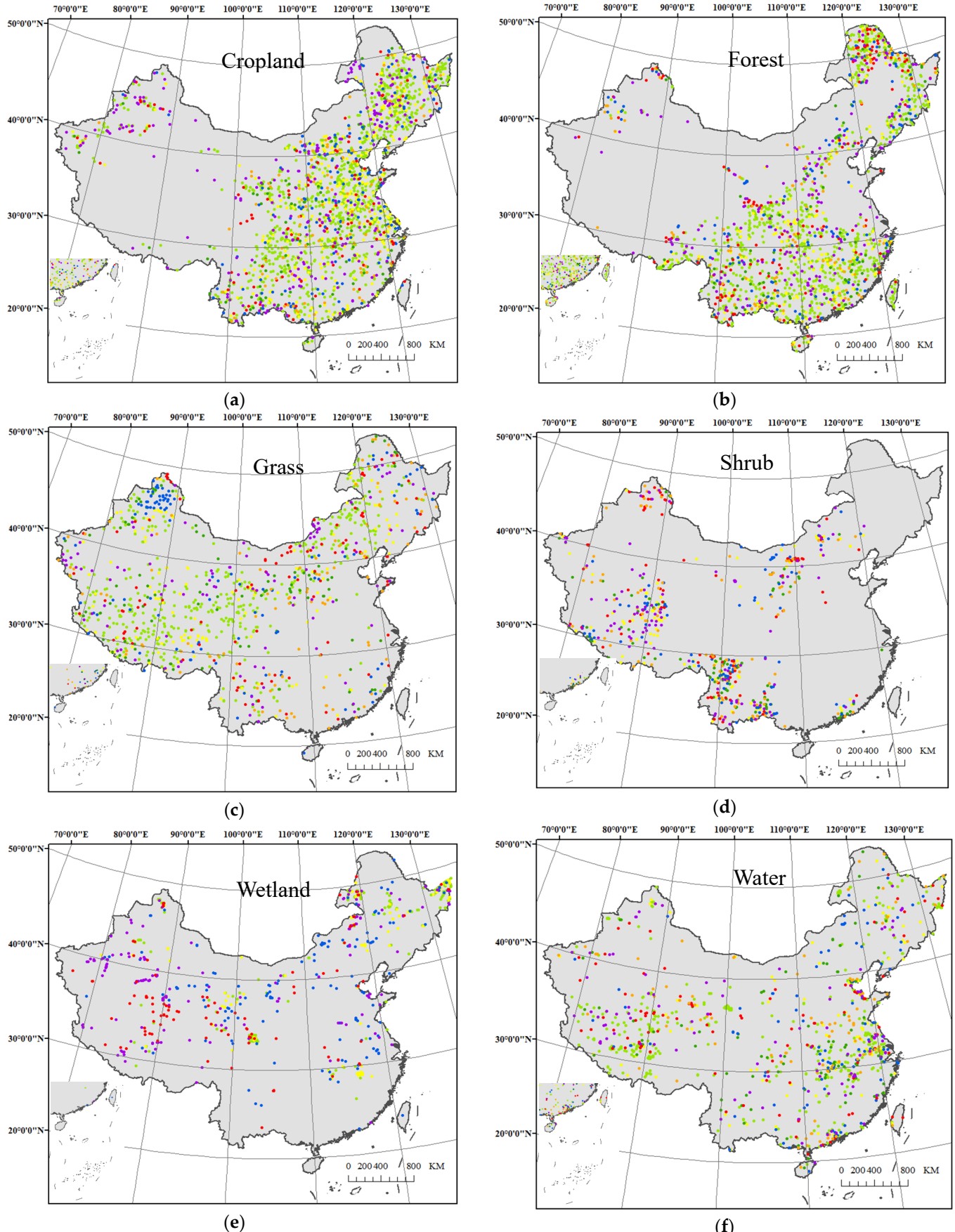

**Figure 5.** *Cont.*

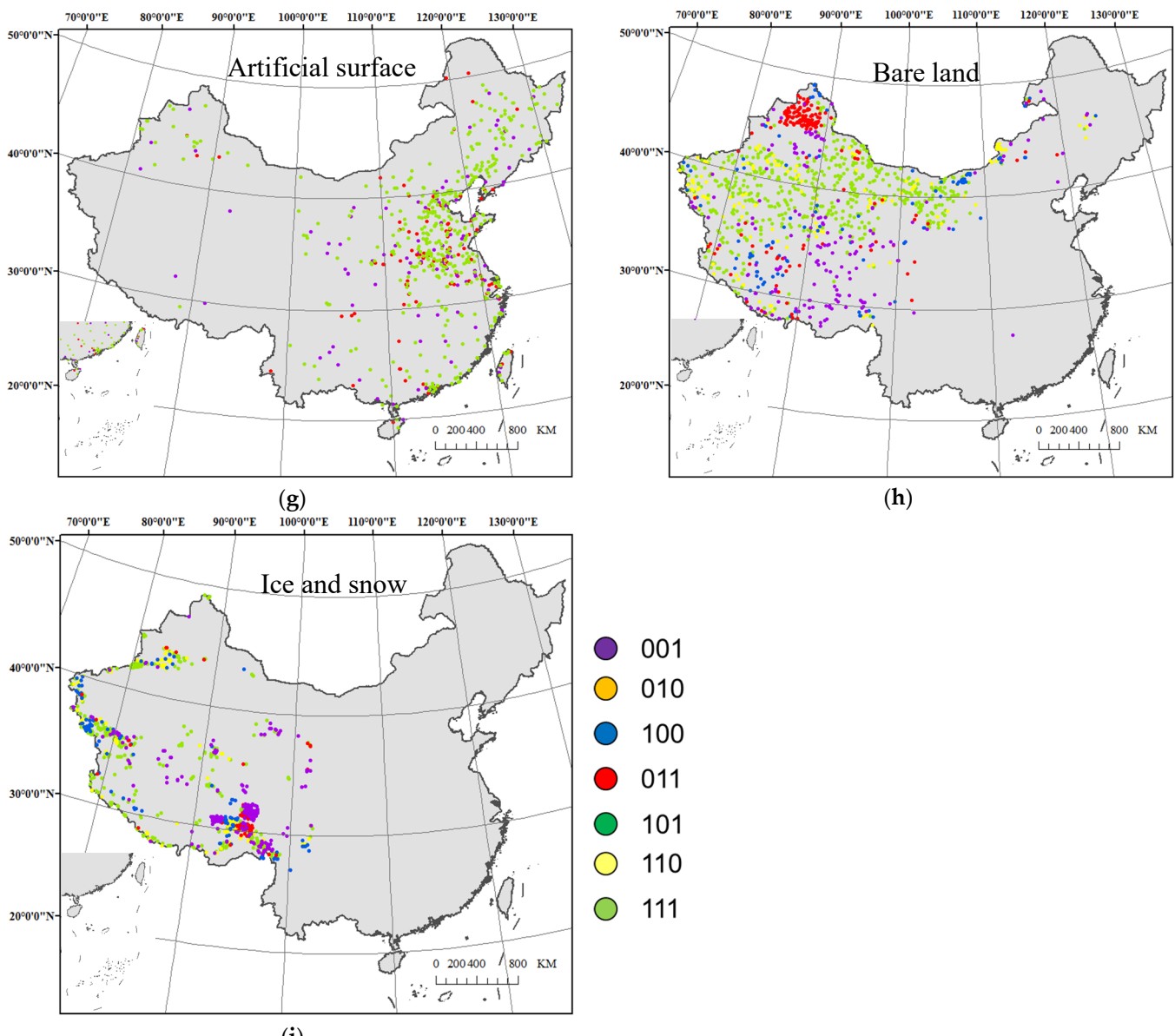

**Figure 5.** Spatial distribution of the samples in each spatiotemporal stratum of each class (001–111 is the binary code of the spatiotemporal stratification for each land cover class corresponding to Table 2). (**a**) Cropland; (**b**) Forest; (**c**) Grass; (**d**) Shrub; (**e**) Wetland; (**f**) Water; (**g**) Artificial surface; (**h**) Bare land; (**i**) Ice and snow.

### 3.2. Multi-Temporal and Specified Class Accuracy

The user accuracy and producer accuracy of the changed and unchanged strata of each class in the three periods are shown in Figure 6. It can be seen from the results (Figure 6) that the accuracy of each class was quite different, but the accuracy of the stable strata (111) of each class was higher, and the user accuracy of this strata was greater than the producer accuracy. However, the user accuracy (UA) of the changed strata was generally poorer than the producer accuracy (PA). The user accuracy of the change strata of the artificial surfaces was higher than that of other classes. The standard error of the producer accuracy for forest, grass, shrub, and bare land was high, and this may be attributed to a large classification error. Generally, the user accuracy of the changed strata was smaller than the producer accuracy, which was the opposite of that of the unchanged strata; thus, the classification error had a greater impact on the changed strata. The user accuracy of the

change in artificial surfaces was higher than that of the other types of change. The high standard error of the mapping accuracy of forest, grassland, shrub, and bare land may be due to the similar spectral values and large classification errors of these categories. As far as the overall accuracy is concerned, the accuracy of bare land was the highest, while that of shrub was the lowest. The reason is that in bare land areas, the area of the unchanged regions was larger, whereas, for the shrub, the area of changed regions was larger, resulting in the classification error having a significant impact on the overall accuracy.

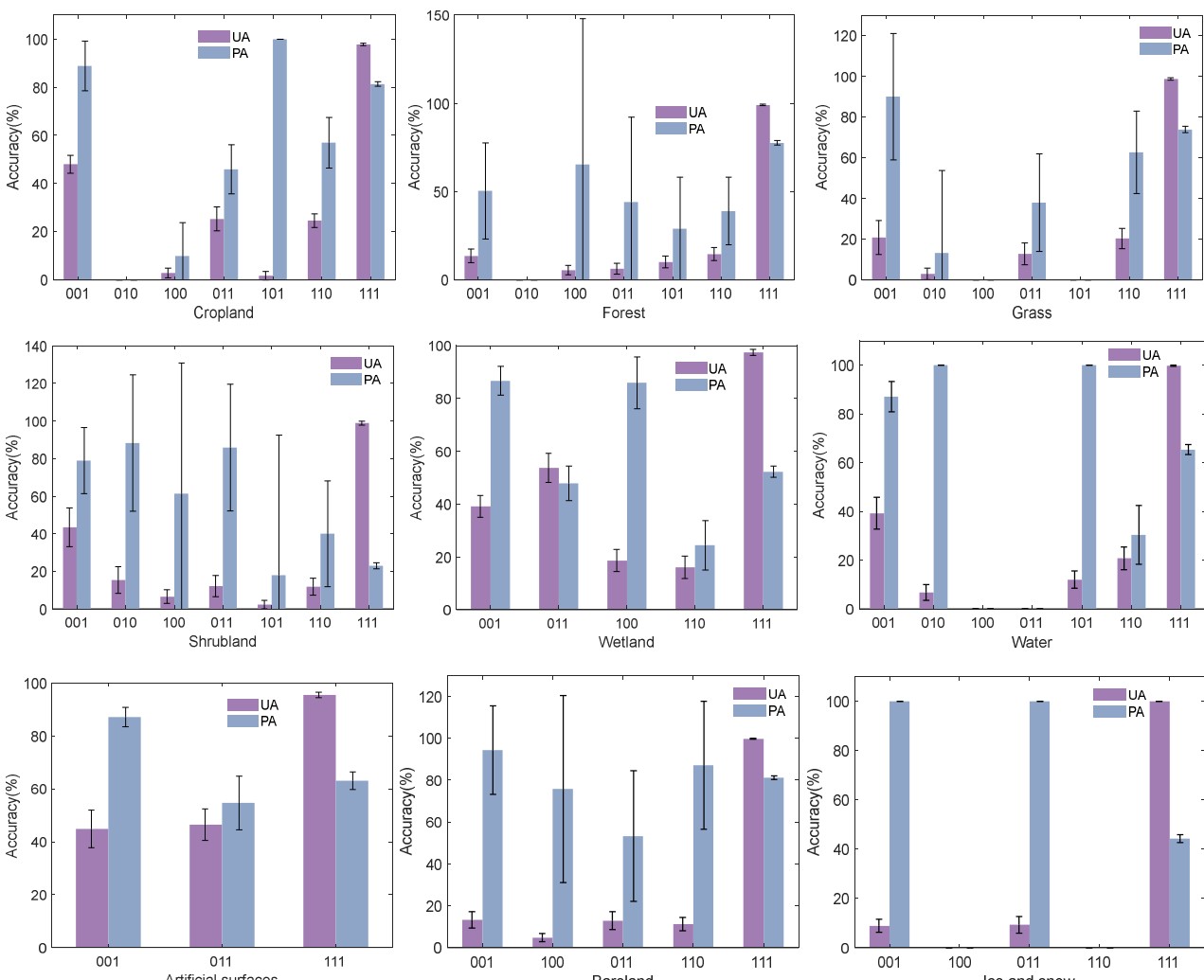

**Figure 6.** User accuracy and producer accuracy of the spatiotemporal stratified sample of each class. The error bars refer to the error of the estimated parameter with 95% confidence intervals. (001–111 is the binary code of the spatiotemporal stratification for each land cover class corresponding to Table 2).

### 3.3. Single Data and Multiclass Accuracy

The accuracy evaluation of the single data and multiple classes was conducted to evaluate the accuracy of the GlobeLand30 land cover data for China in each period. Spatiotemporal stratified sampling provides a variety of possibilities for evaluating the accuracy of single-data and multiclass land cover data. Users can choose the strata according to their needs so as to obtain the desired precision of some stratum or a combination of several strata.

### 3.3.1. The Accuracy of All Strata for Each Class

The accuracy of the period contains all of the strata of each class, as shown in Figure 7. In this case, the sample data of strata 100, 101, 110, and 111 were considered in 2000, the sample data of strata 011, 010, 110, and 111 were considered in 2010, and the sample data of strata 001, 011, 101, and 111 were considered in 2020. The overall accuracy (OA) was 80%, 79%, and 76.2% for 2000, 2010, and 2020, respectively (Figure 7). Since the user accuracy and producer accuracy of shrub and wetland are rather low, the increasing area proportion of these two classes leads to a decreasing overall accuracy over time. The standard error of the OA was less than 1% regardless of the time period. The UA values for cropland, forest, artificial surfaces, bare land, and ice and snow exceeded 80%. The UA and PA for 2000, 2010, and 2020 exceeded 60% for all classes except the UA for shrub. The PA of shrub, water, artificial surfaces, and ice and snow in the three periods is greater than the UA. The UA for cropland, forest, grass, and bare land in the three periods is greater than the PA. However, the standard error of the cartographic accuracy was greater than the standard error of the user accuracy for many categories, such as the water, artificial surfaces, and ice and snow regions, suggesting that the classifiers used to produce data products performed poorly in these land cover classes.

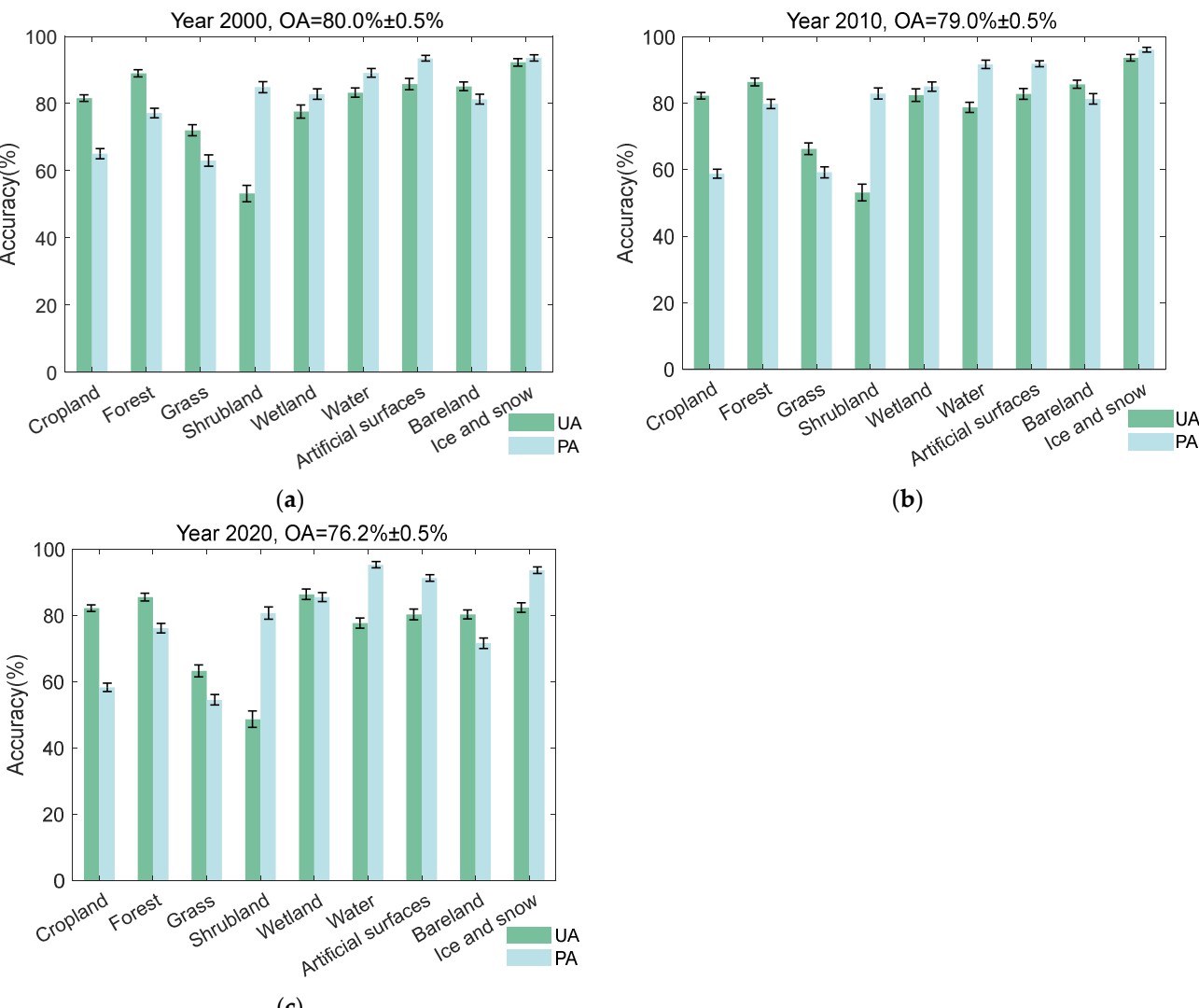

**Figure 7.** The accuracy of the period contains all of the strata of each class. (**a**) Year 2000, OA = 80 ± 0.5%. (**b**) Year 2010, OA = 79 ± 0.5%. (**c**) Year 2020, OA = 76.2 ± 0.5%.

### 3.3.2. The Combined Accuracy of the Strata That Changed Once and the Strata That Were Stable and Unchanged

The combined accuracy of the strata that changed once and the strata that were stable and unchanged in each class is shown in Figure 8. In this case, only the sample data of strata 100, 110, and 111 were considered in 2000, the sample data of strata 111 and 110 were considered in 2010, and the sample data of strata 001, 011, and 111 were considered in 2020. The overall accuracy (OA) was 81.5%, 81.5%, and 77.4% for 2000, 2010, and 2020, respectively (Figure 8). The user accuracy for grass, shrub, water, artificial surface, and ice and snow in 2020 was lower, resulting in an overall accuracy that was lower than that of 2000 and 2010.

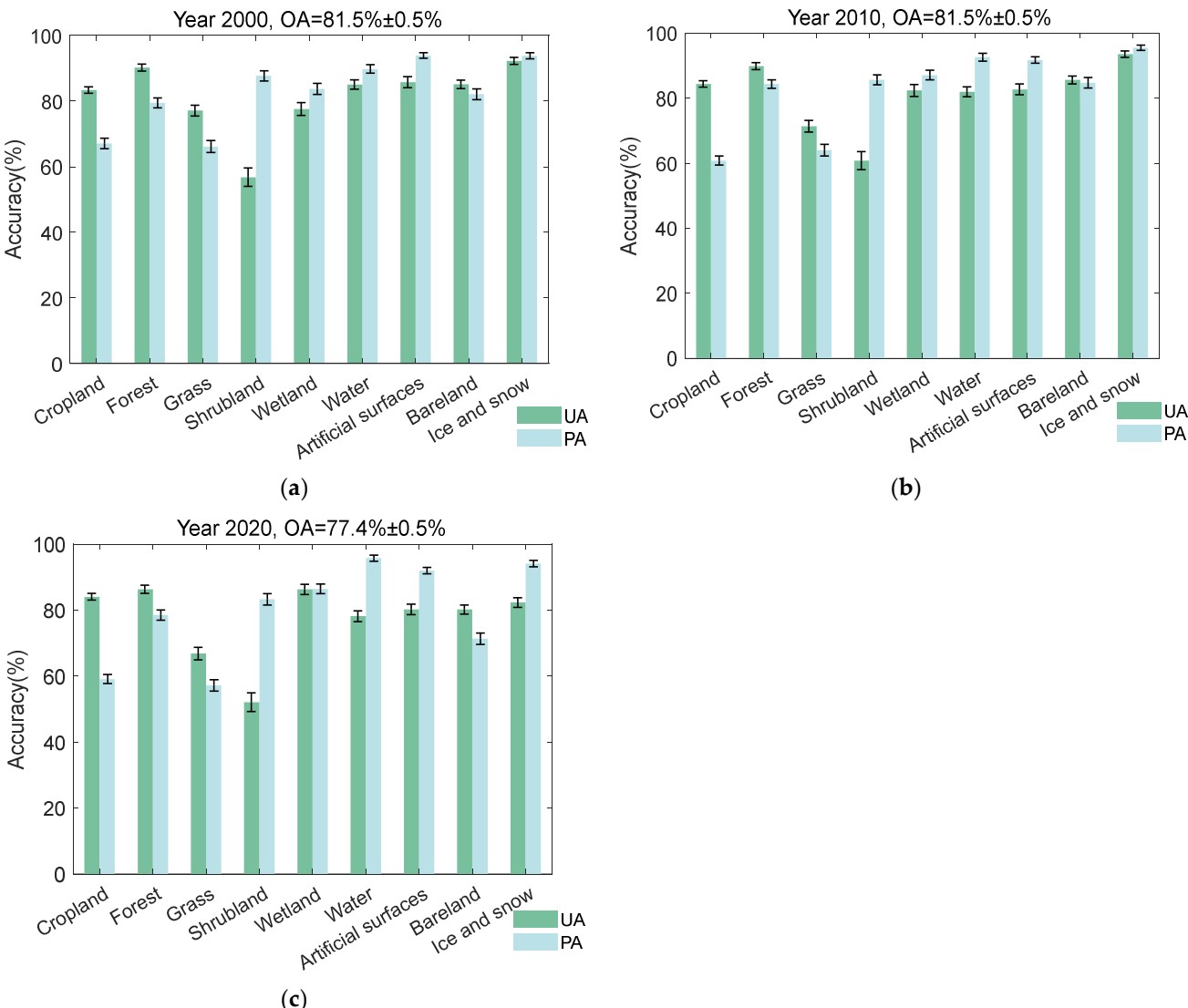

**Figure 8.** The combined accuracy of the strata that changed once and the strata that were stable and unchanged. (**a**) Year 2000, OA = 81.5 ± 0.5%. (**b**) Year 2010, OA = 81.5 ± 0.5%. (**c**) Year 2020, OA = 77.4 ± 0.5%.

### 3.3.3. The Combined Accuracy of the Three Periods as Stable and Unchanged Strata

The accuracy of the three periods in terms of the stable and unchanged strata of each class is shown in Figure 9. In this case, only the sample data of stratum 111 were considered in 2000, 2010, and 2020 for each class. The overall accuracy of the three periods was higher than 90%. The user accuracy of each class exceeded 80%, and the ice and snow area had the highest user accuracy for the three periods. The overall accuracy of the three phases

of data is very close, which also shows that the 111 strata have a very stable classification accuracy. The accuracy of the unchanged strata was higher than that of the combination of the changed and unchanged strata.

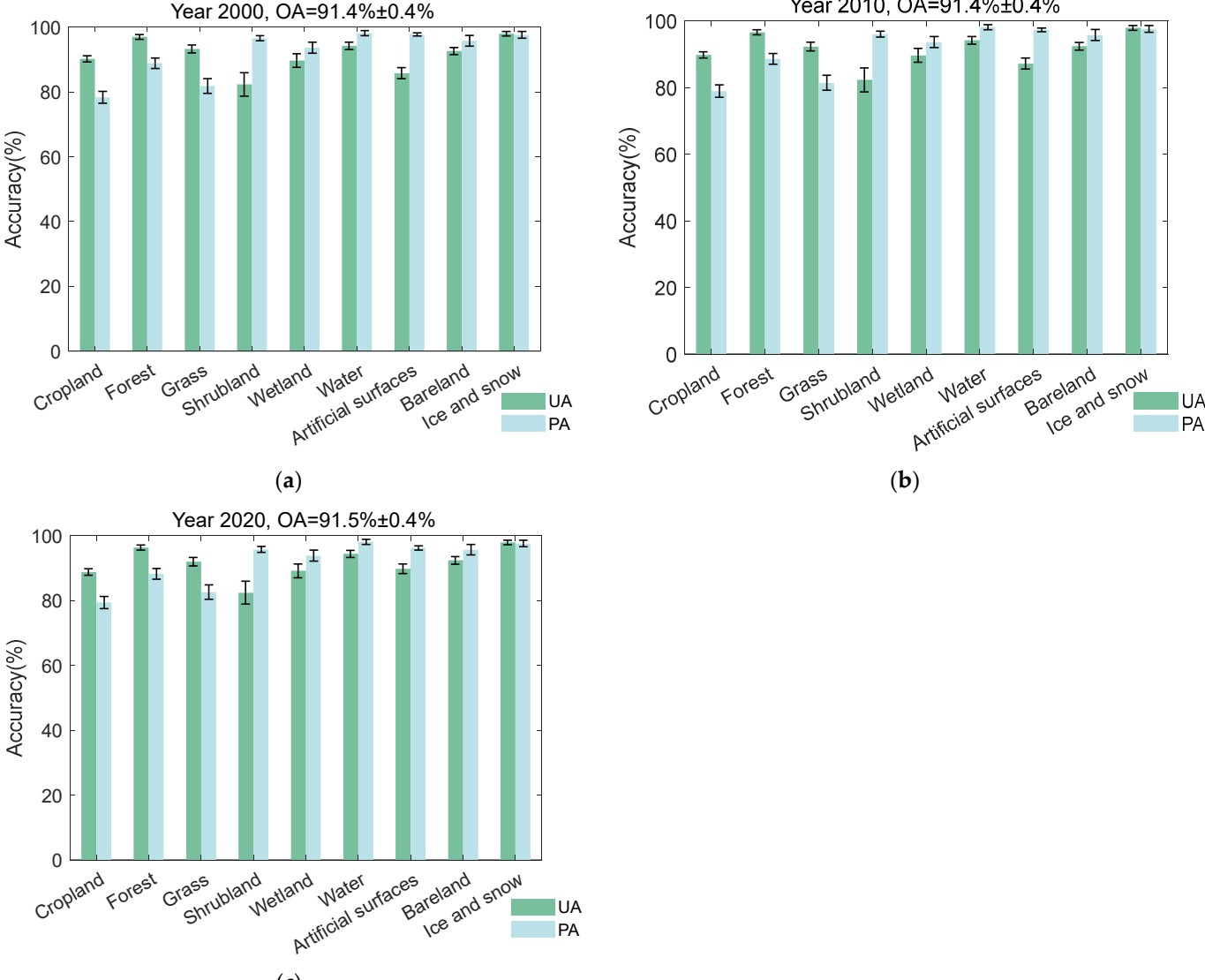

**Figure 9.** The combined accuracy of the three periods according to stable and unchanged strata. (**a**) Year 2000, OA = 91.4 ± 0.4%. (**b**) Year 2010, OA = 91.4 ± 0.4%. (**c**) Year 2020, OA = 91.5 ± 0.4%.

From the different strata of the accuracy results for each class in the three periods (Figures 7–9), the combined accuracy of the strata that changed once and the strata that were stable and unchanged for each class of the GlobeLand30 data for China was close to the result presented by Xie et al. (80.46%) [42] and the result reported by Wang et al. (84.2%) [43]. Stratified sampling based on the category of land cover fails to fully consider the samples of the changed strata, which is an important reason for the difference in accuracy between them. In the case of little difference between sample sizes, when considering the combination of different strata, the accuracy is different. Setting different strata has different significance effects on accuracy. However, it is certain that strata with a higher change frequency will have a lower accuracy. If these strata are selected as a single stratum, the number and weight of the unreliable samples will undoubtedly be increased, and the impact on the overall accuracy will also be evident. When compared to the single data stratified sampling method by land cover class, the spatiotemporal stratified sampling of

multi-temporal and multiclass land cover data showed that classification errors were more likely to occur in the strata that rarely changed. As such, how should we consider the weight of such errors when calculating the accuracy of data products?

### 3.4. Binary Change and No-Change Accuracy

Based on the results of the temporal and spatial stratification of multi-temporal land cover data, the accuracy of the binary changed and unchanged data was evaluated. For example, the class-specific changes from 2000 to 2010 were defined as a nonspecific category in 2000 and converted into a specific category in 2010, and the specific category in 2000 was converted to a nonspecific category in 2010. The 2000–2010 specific categories that were unchanged were defined, as per the 2000–2010 period—they were all specific categories. The category-specific change for 2010–2020 was defined as 2010 being a nonspecific category that was converted into a specific category in 2020, and 2010 was a specific category that was converted into a nonspecific category in 2020. Furthermore, the 2010–2020 specific categories that were unchanged were defined as a specific category for 2010–2020. Table 5 shows the overall accuracy, user accuracy, producer accuracy, and accuracy of the unchanged and changed classes of the nine categories during the 2000–2010 and 2010–2020 periods. The accuracy of unchanged forest, changed forest, grassland, and bare land was high, whereas the accuracy of the changed regions was low, but their overall accuracy exceeded 90%. The reason for this was that the changed areas of the land cover types were much smaller than the unchanged areas; thus, the classification error of the unchanged areas had little impact on the overall accuracy. The overall accuracy reflected the advantages of high user accuracy and the accuracy for the unchanged areas. However, when the area proportion of a specific category of changed areas was not that different from that of the unchanged areas, the accuracy of the unchanged areas affected the overall accuracy, e.g., the user accuracy of the unchanged shrub, wetland, artificial surfaces, and ice and snow regions was high. However, the producer accuracy was reduced due to the omission error of change. The standard error of the accuracy of the unchanged areas was smaller than the standard error of the accuracy of the changed areas, which indicates that the data quality of the changed areas was lower than that of the unchanged areas; this also verifies the need for the spatiotemporal stratification of data. The stratification of data for different precisions can improve the reliability of the accuracy evaluation.

**Table 5.** Unchanged and changed user and producer accuracy values expressed as a percentage. Standard error (also as a percentage) in parentheses for the 2000–2010 and 2010–2020 periods.

| Cropland Map | Reference | | | | | | | |
| --- | --- | --- | --- | --- | --- | --- | --- | --- |
| | **2000–2010 OA = 95.1 $\pm$ 0.5%** | | | | **2010–2020 OA = 92.6 $\pm$ 0.6%** | | | |
| | **Unchanged** | **Change** | **Total** | **UA (%)** | **Unchanged** | **Change** | **Total** | **UA (%)** |
| Unchanged | 0.9472 | 0.0095 | 0.9567 | 99 (0.3) | 0.9051 | 0.0139 | 0.919 | 98.5 (0.4) |
| Changed | 0.0394 | 0.0039 | 0.0433 | 9 (1.4) | 0.0598 | 0.0211 | 0.0809 | 26.1 (1.6) |
| Total | 0.9866 | 0.0134 | | | 0.9649 | 0.035 | | |
| PA (%) | 96 (0.2) | 29.1 (9.7) | | | 93.8 (0.3) | 60.3 (5.6) | | |
| Forest Map | Reference | | | | | | | |
| | **2000–2010 OA = 94.79 $\pm$ 0.7%** | | | | **2010–2020 OA = 93.5 $\pm$ 0.7%** | | | |
| | Unchanged | Changed | Total | UA | Unchanged | Changed | Total | UA |
| Unchanged | 0.9416 | 0.0038 | 0.9454 | 99.6 (0.2) | 0.9240 | 0.0082 | 0.9322 | 99.1 (0.4) |
| Changed | 0.0483 | 0.0063 | 0.0546 | 11.5 (1.6) | 0.0573 | 0.0105 | 0.0678 | 15.4 (1.7) |
| Total | 0.9899 | 0.0101 | | | 0.9813 | 0.0187 | | |
| PA | 95.1 (0.2) | 62.2 (14.7) | | | 94.2 (0.3) | 55.9 (10.2) | | |

**Table 5.** *Cont.*

| Grass Map | Reference | | | | | | | |
|---|---|---|---|---|---|---|---|---|
| | 2000–2010 OA = 92.42 ± 0.9% | | | | 2010–2020 OA = 92.4 ± 0.9% | | | |
| | Unchanged | Changed | Total | UA | Unchanged | Changed | Total | UA |
| Unchanged | 0.9217 | 0.0095 | 0.9312 | 99 (0.4) | 0.9142 | 0.0025 | 0.9167 | 99.7 (0.2) |
| Changed | 0.0663 | 0.0025 | 0.0688 | 3.7 (0.9) | 0.0735 | 0.0098 | 0.0833 | 11.7 (1.6) |
| Total | 0.988 | 0.012 | | | 0.9877 | 0.0123 | | |
| PA | 93.3 (0.3) | 21.1 (19.8) | | | 92.6 (0.4) | 79.6 (12.5) | | |

| Shrub Map | Reference | | | | | | | |
|---|---|---|---|---|---|---|---|---|
| | 2000–2010 OA = 60.2 ± 2% | | | | 2010–2020 OA = 57.1 ± 2% | | | |
| | Unchanged | Changed | Total | UA | Unchanged | Changed | Total | UA |
| Unchanged | 0.5780 | 0 | 0.578 | 1 (0) | 0.5311 | 0.0190 | 0.5501 | 96.5 (1.2) |
| Changed | 0.3983 | 0.0237 | 0.422 | 5.6 (1.1) | 0.4096 | 0.0402 | 0.4498 | 8.9 (1.4) |
| Total | 0.9763 | 0.0237 | | | 0.9407 | 0.0592 | | |
| PA | 59.2 (1.3) | 1 (0) | | | 56.5 (1.2) | 67.9 (12.1) | | |

| Wetland Map | Reference | | | | | | | |
|---|---|---|---|---|---|---|---|---|
| | 2000–2010 OA = 74.29 ± 1.9% | | | | 2010–2020 OA = 69.3 ± 1.9% | | | |
| | Unchanged | Changed | Total | UA | Unchanged | Changed | Total | UA |
| Unchanged | 0.6562 | 0.003 | 0.6157 | 99.5 (0.4) | 0.547 | 0.0187 | 0.5657 | 96.7 (1) |
| Changed | 0.254 | 0.0867 | 0.3843 | 25.4 (2.8) | 0.2886 | 0.1458 | 0.4344 | 33.6 (2.7) |
| Total | 0.9407 | 0.0593 | | | 0.8356 | 0.1645 | | |
| PA | 72.1(1.4) | 96.6 (2.3) | | | 865.5 (1.7) | 88.7 (2.9) | | |

| Water Map | Reference | | | | | | | |
|---|---|---|---|---|---|---|---|---|
| | 2000–2010 OA = 88.7 ± 1% | | | | 2010–2020 OA = 86.7 ± 1.1% | | | |
| | Unchanged | Changed | Total | UA | Unchanged | Changed | Total | UA |
| Unchanged | 0.8576 | 0.0019 | 0.8595 | 99.8 (0.2) | 0.8304 | 0.0073 | 0.8377 | 99.1 (0.4) |
| Changed | 0.1115 | 0.0290 | 0.1405 | 20.7 (2) | 0.1254 | 0.0368 | 0.1622 | 22.7 (2.1) |
| Total | 0.9691 | 0.0309 | | | 0.9558 | 0.0441 | | |
| PA | 88.5 (0.5) | 94 (4) | | | 86.9 (0.6) | 83.4 (5.7) | | |

| Artificial surfaces Map | Reference | | | | | | | |
|---|---|---|---|---|---|---|---|---|
| | 2000–2010 OA = 82 ± 1.7% | | | | 2010–2020 OA = 60.5 ± 2% | | | |
| | Unchanged | Changed | Total | UA | Unchanged | Changed | Total | UA |
| Unchanged | 0.7333 | 0.0106 | 0.7439 | 98.6 (0.6) | 0.4642 | 0.0141 | 0.4783 | 97.1 (0.7) |
| Changed | 0.1691 | 0.0871 | 0.2562 | 34 (4.7) | 0.3809 | 0.1409 | 0.5218 | 27 (4.4) |
| Total | 0.9024 | 0.0977 | | | 0.8451 | 0.155 | | |
| PA | 81.3 (1.7) | 89.2 (4.2) | | | 54.9 (2.3) | 90.9 (2.8) | | |

| Bare land Map | Reference | | | | | | | |
|---|---|---|---|---|---|---|---|---|
| | 2000–2010 OA = 93 ± 0.8% | | | | 2010–2020 OA = 92 ± 0.9% | | | |
| | Unchanged | Changed | Total | UA | Unchanged | Changed | Total | UA |
| Unchanged | 0.9256 | 0.0021 | 0.9277 | 99.8 (0.2) | 0.9129 | 0.0005 | 0.9167 | 99.9 (0.1) |
| Changed | 0.0679 | 0.0044 | 0.0723 | 6.1 (1.5) | 0.0793 | 0.0073 | 0.0833 | 8.5 (1.6) |
| Total | 0.988 | 0.012 | | | 0.9877 | 0.0123 | | |
| PA | 93.2 (0.3) | 67.4 (23.2) | | | 92 (0.4) | 93.2 (9.2) | | |

| Ice and snow Map | Reference | | | | | | | |
|---|---|---|---|---|---|---|---|---|
| | 2000–2010 OA = 87.7 ± 1.3% | | | | 2010–2020 OA = 58.7 ± 1.7% | | | |
| | Unchanged | Changed | Total | UA | Unchanged | Changed | Total | UA |
| Unchanged | 0.8724 | 0 | 0.6157 | 1 (0) | 0.5709 | 0.0002 | 0.7818 | 99.9 (0.1) |
| Changed | 0.1228 | 0.0047 | 0.3843 | 3.7 (1.3) | 0.4127 | 0.0163 | 0.2182 | 3.8 (1.1) |
| Total | 0.9952 | 0.0047 | | | 0.941 | 0.059 | | |
| PA | 87.7 (0.8) | 1 (0) | | | 58 (1.5) | 99 (2.6) | | |

### 3.5. Specific Class Binary Change Area Estimation

Estimating the area of each land cover class and the area of change in different phases can be used to evaluate the changed and unchanged information in the land cover data. A specific class net change was defined as the value that increased or decreased in that class over two periods. Between 2000 and 2020, the net increase in the forest area was only slight, and the increase in the forest area between the 2000–2010 and 2010–2020 periods was significant, which is inseparable from the continuous large-scale national afforestation policy in China. However, the loss of forest is related to poor land plowing, conversion to

pasture, and urbanization expansion. In particular, urban expansion led to the intensive conversion of forest regions (see Figure 10). The area of forests, water bodies, artificial surfaces, and bare land saw a net increase over the 20-year period. The increase in forests and the decrease in cropland were partly due to the policy of returning farmland to forest. In addition, there may be overlapping parts between the different types of land cover, such as the mixed pixels for the cropland and forest boundaries, which have a high probability of being misclassified in these two categories, thus affecting the accuracy of the area estimation. In 2020, the area of artificial surfaces in China increased by 56% compared with that in 2000, and the growth rate from 2010 to 2020 was greater than that from 2000 to 2010. In the past 20 years, the water areas of China have increased continuously due to the expansion of lakes and dam construction on the Tibetan Plateau. Generally, ice and snow regions exist in mountain areas with large temperature differences between day and night in the form of mountain and valley glaciers and snow caps. There was a significant difference between the increase and decrease in ice and snow areas in the 2000–2010 and 2010–2020 periods, and the area of permanent snow and ice increased from 2010–2020. From 2000 to 2020, the areas of grass gradually decreased, shrub first increased and then decreased, and the wetland areas gradually increased.

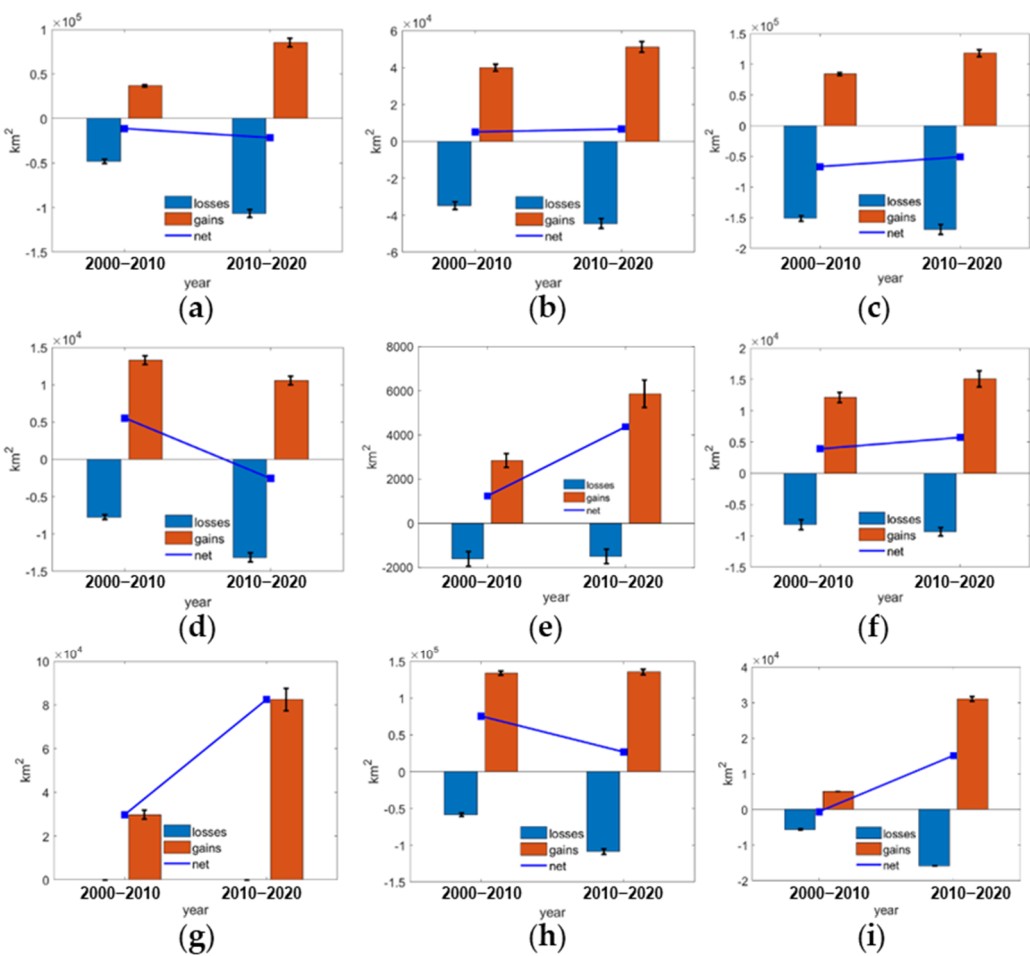

**Figure 10.** Binary data accuracy assessment results for the area estimates of gain, losses, and net change for each land cover class. (**a**) Cropland. (**b**) Forest. (**c**) Grass. (**d**) Shrub. (**e**) Wetland. (**f**) Water. (**g**) Artificial surfaces. (**h**) Bare land. (**i**) Ice and snow.

## 4. Conclusions

The quality control of multi-temporal land cover data with multiple classes is the foundation of many qualitative applications that require the appropriate stratified sampling. Therefore, a spatiotemporal stratified sampling method is proposed by considering both

the changed and unchanged types of each class of data in the three periods. Then, a binary coding approach is utilized to deal with the problem of complex transformation between multiple classes. Additionally, the stratified sample size is allocated by combining the construction area ratio and the minimal value, which ensures enough samples in the changed strata, improving the reliability of the accuracy assessment. The stratified sampling method based on spatiotemporal change can be applied to multi-temporal data with more epochs (the number of time-series datasets is not less than 3). The sampling design is similar to the spatiotemporal stratified sampling method proposed in this paper. However, since more periods of data have more complex transformations, we need to analyze them according to specific data. Refined sample allocation enables an accurate area estimation for land cover changes. Additionally, the samples with frequent changes have a higher probability of returning a classification error, which shows that stratification sampling based on temporal changes is necessary. This experiment, based on the multi-temporal GlobeLand30 land cover data with multiple classes in China, shows that the accuracy of the changed strata was lower than that of the unchanged strata. Since the uncertainty between the two acquisitions was larger than a single one, the accuracy of the multi-temporal land cover data in a single period was higher. The accuracy of the strata that changed more frequently was lower than that of the strata that changed with less frequency. The different classes of land cover showed a difference in user and producer accuracy. When considering the land cover changes in the past 20 years, the results showed that the building, forest, water, wetland, and bare land areas increased significantly, whereas the other types of land cover decreased.

**Author Contributions:** Conceptualization, Y.G., H.X. and X.T.; methodology, Y.G., H.X. and Y.J.; software, Y.G.; validation, Y.G., S.L. and Y.L.; formal analysis, Y.G., C.W. and H.X.; writing—original draft preparation, Y.G.; writing—review and editing, H.X. and X.T.; supervision, X.T.; project administration, H.X.; funding acquisition, H.X. and X.T. All authors have read and agreed to the published version of the manuscript.

**Funding:** This research was funded by the National Natural Science Foundation of China (Project Nos. 42221002 and 41631178), the Shanghai Academic Research Leader Program (23XD1404100), and the Fundamental Research Funds for the Central Universities of China.

**Data Availability Statement:** The data presented in this study are available at globeland30.org.

**Conflicts of Interest:** The authors declare no conflict of interest.

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
