# Peer review of "Assessing the Accuracy of Multi-Temporal GlobeLand30 Products in China Using a Spatiotemporal Stratified Sampling Method"

_remotesensing, doi:10.3390/rs15184593_

Round 1

Reviewer 1 Report

This paper proposes a spatial–temporal stratified sampling method for stratifying the multi-temporal and multi-class Globeland30 products in China. The paper is well-designed and written, and represents a good case study. My major concern is related to the figure quality. The figures are very vague and the sub-figures of the nine-dash line are not complete.

Minor editing of English language required

Reviewer 2 Report

The comments are in the attached file. 

Round 2

Reviewer 3 Report

Dear Authors,

Thank you for your response to my opinion and the related revisions. I have reviewed them, and finally, it would be beneficial to move the 'main contribution' mentioned in lines 124-131 to the conclusion and present it together with the conclusion. Please make this revision.

Best regards,
